# Sex-Related Differences in Post-Stroke Anxiety, Depression and Quality of Life in a Cohort of Smokers

**DOI:** 10.3390/brainsci14060521

**Published:** 2024-05-21

**Authors:** Rosa Suñer-Soler, Eduard Maldonado, Joana Rodrigo-Gil, Silvia Font-Mayolas, Maria Eugenia Gras, Mikel Terceño, Yolanda Silva, Joaquín Serena, Armand Grau-Martín

**Affiliations:** 1Research Group Health and Health Care, Nursing Department, University of Girona, 17003 Girona, Spain; 2Nursing Department, University of Vic, 08242 Manresa, Spain; emaldonado@umanresa.cat; 3Department of Neurology, Dr. Josep Trueta University Hospital, 17007 Girona, Spain; jrodrigo.girona.ics@gencat.cat (J.R.-G.); mikelterceno@hotmail.com (M.T.); ysilva.girona.ics@gencat.cat (Y.S.); jserena.girona.ics@gencat.cat (J.S.); 4Quality of Life Research Institute, University of Girona, 17003 Girona, Spain; silvia.font@udg.edu (S.F.-M.); eugenia.gras@udg.edu (M.E.G.); 5Hospital of Figueres, Fundació Salut Empordà, 17600 Figueres, Spain; grauma@comg.cat

**Keywords:** stroke, smoking, anxiety, depression, quality of life

## Abstract

Background: We aimed to study anxiety, depression and quality of life in smokers after stroke by sex. Methods: A longitudinal prospective study with a 24-month follow-up of acute stroke patients who were previously active smokers. Anxiety and depression were evaluated with the Hospital Anxiety and Depression scale, and quality of life was evaluated with the EQ-5D questionnaire. Results: One hundred and eighty patients participated (79.4% men); their mean age was 57.6 years. Anxiety was most prevalent at 3 months (18.9% in men and 40.5% in women) and depression at 12 months (17.9% in men and 27% in women). The worst perceived health occurred at 24 months (EQ-VAS 67.5 in men and 65.1 in women), which was associated with depression (*p* < 0.001) and Rankin Scale was worse in men (*p* < 0.001) and depression in women (*p* < 0.001). Continued tobacco use was associated with worse perceived health at 3 months in men (*p* = 0.034) and at 12 months in both sexes. Predictor variables of worse perceived health at 24 months remaining at 3 and 12 months were tobacco use in men and neurological damage in women. Conclusion: Differences by sex are observed in the prevalence of anxiety and depression and associated factors and in the predictive factors of perceived health.

## 1. Introduction

Stroke is the second most common cause of death and one of the main causes of incapacity in the world, significantly impacting the quality of life of people who suffer from it [1]. The predictive factors related to quality of life in people with stroke include sociodemographic factors, such as age, biological sex, educational level and socioeconomic level [2]; factors related to the disease itself, such as the severity, physical function and mental state; environmental factors, such as social support [2]; and individual factors, such as coping strategies and self-esteem after stroke [3].

With regard to mental state, anxiety is a psychopathological disorder that is frequent after stroke. When this was evaluated between 0 and 2 week after stroke, it was found in 36.7% of patients, and its prevalence diminished in the first 3 months after stroke to 24.1%, remained stable during the first year (23.8%) and may be a predictive factor of depression [4]. It has been observed that anxiety is more frequent among young people with a history of anxiety or depression and with greater dependency after stroke, associated with a worse quality of life [5]. Having symptoms of anxiety adversely affects rehabilitation and negatively impacts the long-term results and quality of life of patients, making it fundamentally important that it should be diagnosed [6]. Post-stroke depression is the most frequent and least diagnosed mental disorder in people who have previously suffered a stroke, and it has a significant impact on the quality of life, functional recovery, survival and cognitive function of stroke sufferers [7,8,9,10], since they involve biological, behavioural and social aetiopathogenic mechanisms [11,12]. Women after stroke have shown worse health-related quality of life [13,14] and more anxiety [14].Those people who continue smoking have a worse quality of life than those who stop smoking, although this phenomenon is less studied in stroke survivors than in the general population [15,16].

The objectives of the present study were as follows: (1) to find out the prevalence of anxiety and depression in smokers who were stroke survivors and the associated factors; (2) to study changes in anxiety and depression over a two-year period in this cohort with a focus on differences by biological sex; and (3) to evaluate the quality of life in the first two years post-stroke in this cohort and the predictive factors in both sexes.

## 2. Methods

### 2.1. Design

A longitudinal prospective study with follow-up over 24 months of a cohort of patients with acute stroke who were smokers before having the stroke.

### 2.2. Participants

A cohort of patients who were active smokers consecutively admitted to the Dr. Josep Trueta University Hospital in Girona with an acute stroke diagnosis (ischaemic or haemorrhagic). Patients who were in a life-threatening condition (Rankin Scale ≥ 4) or who suffered from aphasia were excluded.

### 2.3. Measures

Sociodemographic variables (age, sex, education level and living with a partner), the presence of vascular risk factors (high blood pressure, diabetes, dyslipidaemia, embolic heart disease), as well as tobacco and other addictions were all studied. The characteristics of the neurological lesion were also studied.

All patients included in the study received either a brief anti-smoking intervention or an intensive intervention consisting of health education together with a motivational interview, which was structured depending on the stage of change with regard to their smoking habit, within a clinical trial setting [17].

During the follow-up, the functional situation was analysed at 3 months, 12 months and 24 months using the modified Rankin Scale [18]; neurological status was analysed with the National Institutes of Health Stroke Scale (NIHSS) [19]; and smoking status was measured by self-declaration and objective CO-oximetry assessment, taking a value of less than 8 ppm of CO as confirmation of being a non-smoker (had there been any discrepancy between the two, we would have taken the objective measure as defining their smoking status) [20].

Anxiety and depression were assessed using the Hospital Anxiety and Depression scale (HAD scale) [21,22]. The HADS is a questionnaire with 14 statements (7 to assess anxiety and 7 for depression), with scores ranging from 0 to 3, with possible values of 0 to 21 for each of the subscales. This scale measures status as it explores the situation over the previous week. Designed to be administered to subjects with organic diseases, the physical aspects that can accompany anxiety or depression have been eliminated so as to focus only on the emotional ones [21,22,23,24,25]. The authors consider that an upper limit with values of 10 or 11 should be used for research purposes in order to obtain a low proportion of false positives whereas values of 8 or 9 would avoid false negatives [22]. For this study, a cut-off value of 10 was taken for the presence or absence of anxiety and depression.

Quality of life related to perceived health was studied using the EuroQol-5D questionnaire (EQ-5D), validated for Spanish populations. In the first part of the questionnaire, five dimensions are evaluated: mobility, self-care, usual activities, pain/discomfort and anxiety/depression. Responses for the five dimensions can be combined into a five-digit number that describes the interviewees’ health state and this can be converted into a single utility index. The second part of the EQ-5D is a 20-centimetre vertical visual analogue scale (EQ-VAS) from 0 (worst imaginable health state) to (best imaginable health state) [26,27].

### 2.4. Follow-Up of Participants

Once the project for the study had been approved by the Ethical and Clinical Trials Committee, participants admitted to the Neurology Service with a diagnosis of acute stroke began to be included. Recruitment was between 2015 and 2018, and follow-up was performed at 3, 12 and 24 months after the stroke at the Outpatients Department of the same hospital, coinciding with scheduled appointments with the neurologist. In order to minimize losses to follow up, the investigators conducted the visit at the participants’ homes if they did not attend the hospital for the programmed visit.

### 2.5. Ethics Statement

This study has been conducted respecting the Helsinki Declaration and the rules of Good Clinical Practice. The confidentiality of information and anonymity was guaranteed at all times. All patients were given detailed information about the objectives and nature of the study both orally and in writing and all participants gave their written informed consent. The study was approved by the Ethics Committee of the Girona Biomedical Research Institute (Spain).

### 2.6. Statistics

The statistical analysis was performed using SPSS for Windows software, version 25.0 (IBM). Categorical variables are described as absolute frequencies and percentages and continuous variables are described as means (standard deviation) or medians (25 and 75 percentiles). The chi-squared test was used to compare categorical variables, Student’s *t* test or the Mann–Whitney U test to compare independent groups and Student’s *t* test to compare related groups for continuous variables.

In order to establish the variables associated with the Perceived Health dependent variable of the VAS at 3 months, 12 months and at 24 months after stroke, separate multivariable studies were performed for men and women using multiple linear regression with the forward method to determine the independent variables to use in the model from the following options: age; educational level; living with a partner; number of cardiovascular risk factors accompanying tobacco addiction; presence of other addictions; type of neurological lesion; type of intervention; and Rankin Scale scores, NIHSS scores, pain/discomfort (EQ-5D subscale), HAD Anxiety Scale scores, HAD Depression Scale scores and evaluation of the state of tobacco addiction (active smoker or not) at 3, 12 or 24 months, depending on the dependent variable being studied.

In order to establish the predictor variables of the Perceived Health dependent variable 24 months after stroke, a multivariable study was performed between the evaluations registered at 3 and 12 months after stroke for men and women through multiple linear regression with forward selection to determine the independent variables to include in the model from the following items: age; educational level; living with a partner; number of cardiovascular risk factors together with the smoking habit; presence of other addictions (in addition to tobacco); type of neurological lesion; type of intervention; and Rankin Scale scores, NIHSS scores, pain/discomfort (EQ-5D subscale), HAD Anxiety Scale scores, HAD Depression Scale scores and evaluation of the state of the smoking habit (active smoker or not) at 3 and 12 months.

## 3. Results

One hundred and eighty participants (79.4% males with a mean age of 57.6 years) were studied 3 months after hospitalization for stroke. Between 3 months and 12 months two participants died and seven were lost to follow-up. A further five died between 12 and 24 months and two more participants were lost. All were active smokers until having the stroke and none smoked when hospitalized, during which time they received either brief antismoking advice or a more intensive structured motivational intervention. The characteristics of the sample are described in Table 1.

The prevalence of anxiety at 3 months was 23.3% (42 cases). In comparison with the participants with HAD Anxiety Scale scores < 10 points, people with anxiety were younger, more often women, less often living together with partners and more often active smokers due to having relapsed and scored higher on the HAD Depression Scale (Table 1). The prevalence of depression in the whole sample was 11.1% (20 cases). In comparison with the participants with HAD Depression Scale scores < 10 points, participants with depression were more often consumers of other addictive substances (in addition to tobacco) before stroke, had a worse functional state 3 months after stroke (Rankin Scale) and scored higher on the HAD Anxiety Scale (Table 1).

Twelve months after stroke, the prevalence of anxiety was 19.3% (33 cases) whereas the prevalence of depression was 19.9% (34 cases).

Twenty-four months after stroke, the prevalence of anxiety was 20.7% (34 cases) and the prevalence of depression was 15.9% (26 cases). People with anxiety at 24 months were younger (*p* < 0.001), predominantly women (*p* = 0.004), proportionally more often smokers (*p* = 0.017), and proportionally less often living with a partner (*p* = 0.045) and had higher scores on the HAD Depression Scale (*p* < 0.001). People with depression had greater neurological deficits (*p* = 0.001), greater functional deficits (*p* < 0.001) and higher scores on the HAD Anxiety Scale (*p* < 0.001) (Table 2).

## 4. Evolution of Anxiety and Depression in Men and Women

The prevalence of anxiety in men between 3 and 12 months fell from 18.9% to 14.2% and at 24 months it was 15.7%. In women this prevalence fell from 40.5% to 37.8% at 12 months and this was maintained at 24 months. The mean score on the HAD Anxiety Scale did not show significant variations between 3 and 12 months in men (mean score 4.9 vs. 5; *p* = 0.91) or in women (7.5 vs. 8.5; *p* = 0.19). The mean score on the HAD Anxiety Scale did not show significant variations either between 12 and 24 months in men (5 vs. 4.8; *p* = 0.51) or women (8.5 vs. 7.5; *p* = 0.09).

The prevalence of depression in men increased from 11.2% at 3 months to 17.9% at 12 months and decreased to 16.5% at 24 months. In women the prevalence increased from 10.8% at 3 months after stroke to 27% at 12 months and fell to 13.5% at 24 months. The mean score on the HAD Depression Scale showed significant variations between 3 and 12 months in men (4.2 vs. 5.1; *p* = 0.029) and women (4.1 vs. 6.8; *p* < 0.001). Between 12 and 24 months, the mean score on the HAD Depression Scale did not show significant variations in men (5.1 vs. 4.6; *p* = 0.21), but did in the case of women (6.8 vs. 5.4; *p* = 0.008).

## 5. Quality of Life at 3 Months and Associated Factors

Three months after stroke, the mean score for perceived health (EQ-VAS) on a scale from 0 to 100 was 68.5 (18.2). In men, the mean score was 68.2 (18.4), and in women, the mean was 69.9 (17.6). The EQ index gave a median of 0.81 (P25–P75: 0.68–0.91). The median for men was 0.81 (P25–P75: 0.68–0.91), whereas in women it was 0.80 (P25–P75: 0.68–0.91).

In the linear regression model, the factors associated with improved perceived health in men at 3 months were having lower scores on the HAD Depression and Anxiety scales, better functional state according to the Rankin Scale and not having relapsed with regard to smoking in the 3 months since stroke (Table 3). In women, the only associated factor was having a lower score on the HAD Depression Scale (Table 4).

## 6. Quality of Life at 12 Months and Associated Factors

Twelve months after stroke, the mean perceived health on the EQ-VAS scale was 67.5 (20). In men, the mean was 68.1 (19.5), whereas the mean score in women was 65.6 (21.8).

The median EQ index score was 0.82 (P25–P75: 0.68–0.92). The median score in men was 0.83 (P25–P75: 0.68–0.92), and in women it was 0.79 (P25–P75: 0.66–0.89).

The linear regression model found the factors associated with better perceived health in men at 12 months to be a lower score on the HAD Depression Scale, a lower pain score, improved functional state as measured by the Rankin Scale and not smoking (Table 3). In women the associated factors were a lower HAD Depression Scale score and not smoking 12 months after stroke (Table 4).

## 7. Quality of Life at 24 Months and Associated Factors

Twenty-four months after stroke, the mean EQ-VAS scale was 67 (22.2). In men, the mean was 67.5 (21.6), whereas the mean score in women was 65.1 (24.5).

The median EQ index score was 0.85 (P25–P75: 0.72–0.93). The median score for men was 0.85 (P25–P75: 0.71–1) and in women it was 0.85 (P25–P75: 0.77–0.91).

In the linear regression model, the factors associated with better perceived health in men at 24 months were having a lower HAD Depression Scale score and a better functional state as measured by the Rankin Scale (Table 3). In women, the only factor associated was a lower score on the HAD Depression Scale (Table 4).

## 8. Predictor Variables of Better Quality of Life

Finally, the variables that predict better quality of life 24 months after stroke at the 3-month evaluation were lower level of anxiety with abstinence from smoking in men and less neurological deficit in women. The variables predicting better quality of life at 24 months after stroke in the evaluation at 12 months were a lower level of pain, better functional state, living with a partner and abstention from tobacco in men and a lower level of depression and lesser neurological deficits in women (Table 5).

## 9. Discussion

This study has investigated the evolution of anxiety and depression and of quality of life (particularly of perceived health) in a cohort of patients with acute stroke and tobacco use at the time of stroke over a two-year period. The typical profile of the participants was of a young adult, mostly male, with mild neurological damage as a result of the stroke and mild functional deficit. Most patients had at least one other vascular risk factor in addition to tobacco use.

Three months after stroke, approximately one in four participants had anxiety, with a greater frequency among women, and one out of ten participants had depression with similar prevalence in the two sexes. At 12 months, the prevalence of anxiety and depression had practically evened out, affecting two out of ten participants (anxiety decreased and depression increased). At 24 months after stroke, the prevalence of anxiety remained stable and depression decreased.

With regard to anxiety, our results are similar to those described in a systematic review of 97 studies, including 22,262 stroke survivors, with a prevalence of 23.6% in the first five months and 21.5% from 6 to 12 months [28]. Similar figures for the prevalence of anxiety (23.8%) were found in a separate systematic review that included 37 studies evaluating a total of 13,576 patients [4]. In the case of depression, our study obtained prevalence rates lower than those reported by other authors with values of 31% in the first 6 months, in comparison with 33% from 6 to 12 months and 25% from one year [29], or of 28.5% in a meta-analysis with an average follow-up of 6.8 months that included 15,573 people [30].

In order to correctly interpret the comparison of prevalence rates, it is important to take into account the choice of a higher cut-off point in the present study on the HAD scale for cases of anxiety and depression with the aim of achieving greater specificity in comparison with most of the studies revised in the literature, which opted for greater sensitivity with the possibility of more false positives [22].

In the study of factors associated with anxiety during the follow-up, sociodemographic factors such as lower age, female sex and not living with a partner particularly stand out, as do being an active smoker and higher levels of depression. On the other hand, the factors associated with depression were related to the consequences of the stroke itself, such as neurological and functional deficits and higher levels of anxiety. An association has been found between anxiety and depression with greater functional deficit, as well as an association of female sex with anxiety, but not with depression, after stroke [14]. A greater prevalence of depression in cases of left-side stroke (dominant) and in patients with aphasia has been described [30] that has not been observed in the present study, which did not include patients with communication problems.

There were more cases of anxiety in women during evolution and there was less of a reduction in its prevalence in this sex. Unlike the stability described by Sadlonova et al., [14] the prevalence of depression in both sexes increased at 12 months before decreasing at 24 months. This increase in cases during the first year was more pronounced in women.

With regard to quality of life in the two years immediately after stroke, the perception of the participants regarding their health state (EQ-VAS) was lower than the normative values for the Spanish population in the age group of 55 to 64 years, which corresponds to the mean age of our sample. On comparing the EQ-5D index, a value of 0.85 was found in both sexes as compared to normative values of 0.92 in women and practically 1 in men [31]. Xie et al. observed average scores of perceived health (EQ-VAS) that were significantly lower in people with stroke of unknown time of onset in comparison with a general population that had not suffered a stroke, in both men (62.5 vs. 81.5) and women (60.9 vs. 79.5), which were slightly lower than those found in the present study at 24 months (a mean of 67.5 in men and 65.1 in women) [32].

Stability in the population values for perceived quality of life was observed over the two years of follow-up in both sexes, coinciding with the Oxford cohort results from the International Stroke Outcomes Study (INSTRUCT) with median EQ index values at one year and five years of 0.80 in men and 0.73 and 0.71, respectively, in women [33]. An association has been found between worse quality of life after stroke and anxiety, depression [14,34] and female sex [14], as well as functional state [14,35,36]. Although mood disorders are more prevalent in women and there are some differences by sex in the picture of symptoms and the course of the disease, it is difficult to identify biological and cultural factors that contribute to this state of affairs. It has been proposed that processes that might contribute to the increased rate of mood disorders in women are a greater flow of reproductive hormones over the course of their lives and a greater sensitivity to catecholamines. Given this, strategies to clearly identify the mediators of sex differences in mood disorders should include longitudinal observational studies during periods of hormonal changes, as well as psychological and pharmacological clinical trials studying factors that are hypothesised as contributing to the vulnerability of women. None of these variables have been considered in the present study [37,38].

In the present study, the analysis of factors associated with quality of life has been segregated by sex, given the influence that sex is known to have on quality of life [39]. Perceived health is principally associated with depression, functional state and the active consumption of tobacco in men and with depression in women. Other authors have found an association between worse perceived health (EQ-VAS) and a greater level of dependency as well as a higher score on the Hamilton Depression Rating Scale [40]. This association has also been observed studying quality of life with other instruments such as the SF-36 [41].

The most stable predictor of worse perceived health at 24 months was active tobacco use in men and the neurological state (NIHSS) in women. In this respect, the relationship between tobacco addiction and worse quality of life has been clearly demonstrated [15,16,42]. Smoking cessation leads to an improvement of mental health with a decrease in anxiety, depression and mixed pathology [43,44,45] and with improved quality of life [43,44,45]. With regard to the neurological deficit, patients with higher scores on the NIHSS scale reported more problems in all dimensions of the EQ-5D, with the result that the NIHSS score explained 45.8% of the variation in mobility, 54.5% in self-care and 48.9% in daily living activities, although only 7.5% in pain/discomfort and 5.8% in anxiety/depression [46]. A lower quality of life from 6 to 12 months after stroke, measured with the Stroke-Specific Quality of Life Scale (SS-QoL), was independently associated with NIHSS scores >4 at 24 months after stroke [47].

The most important limitations of this study are not knowing the presence of anxiety or depression before stroke, and the lack of pharmacological histories of the participants with regard to mental health issues. As has been mentioned, people with aphasia and those who were institutionalized were excluded due to the need to obtain the stroke patients’ own answers to the questionnaires, as were people with severe neurological deficits. Another limitation is the small number of women who were recruited to the cohort, which is largely due to the low proportion of women affected by stroke in the area we serve [48] and the lower prevalence of tobacco use in women [49].

The strengths of the study include the longitudinal follow-up with face-to-face interviews over a two-year period and having chosen a higher cut-off point in the HAD questionnaire for cases of anxiety and depression in order to obtain greater specificity. A further strength is that greater objectivity is achieved in assessing whether a person has given up smoking or not by measuring the level of exhaled CO. To the best of our knowledge, there are no other follow-up studies of cohorts of active smokers before stroke measuring the evolution of their mental health and quality of life.

The results of the present study show an association between poor perceived health and depression and that anxiety in the first months after stroke and depression during the first year is predictive of worse perceived health, suggesting the importance of identifying and correcting mental health disorders immediately after stroke.

## 10. Conclusions

Stroke survivors showed a clinically significant prevalence of anxiety and depression. Anxiety, which was particularly prevalent among women, is associated with intrinsic factors and being an active smoker after stroke. On the other hand, depression was associated with clinical factors resulting from the stroke, such as neurological and functional deficits. These mental health disorders and the persistence of active smoking had an influence on lower perceived quality of life in patients with mild to moderate post-stroke sequelae.

## Figures and Tables

**Table 1 brainsci-14-00521-t001:** Characteristics of the overall sample 3 months after stroke by diagnosis of anxiety and depression.

	Global (180)	Anxiety (42)	No Anxiety (138)	*p*	Depression (20)	No Depression (160)	*p*
Age	57.6 (11.4)	52.3 (11.5)	59.2 (10.9)	0.01	57.3 (8.9)	57.6 (11.7)	0.90
Sex							
*Male*	143 (79.4)	27 (64.3)	116 (84.1)	0.005	16 (80)	127 (79.4)	0.94
*Female*	37 (20.6)	15 (35.7)	22 (15.9)		4 (20)	33 (20.6)	
Other addictions	49 (27.4)	13 (31.7)	36 (26.1)	0.47	10 (50)	39 (24.5)	0.01
Primary level education	114 (63.3)	29 (69)	85 (61.6)	0.38	16 (80)	98 (61.3)	0.10
Living with a partner	68 (37.8)	9 (21.4)	59 (42.8)	0.013	8 (40)	60 (37.5)	0.82
Ischaemic stroke	155 (86.1)	36 (85.7)	119 (86.2)	0.93	15 (75)	140 (87.5)	0.12
Side							
*Left*	92 (51.4)	20 (47.6)	72 (52.6)	0.41	9 (45)	83 (52.2)	0.60
*Right*	72 (40.2)	20 (47.6)	52 (38)		10 (50)	62 (39)	
*Bilateral*	15 (8.4)	2 (4.8)	13 (9.5)		1 (5)	14 (8.8)	
Insula lesion	22 (12.3)	4 (18.2)	18 (13.1)	0.53	2 (10)	20 (12.6)	0.74
Rankin at 3 months *	1 (1–2)	2 (1–2)	1 (0.75–2)	0.11	1 (1.25–3)	1 (0–2)	0.001
NIHSS at 3 months *	0 (0–1.25)	1 (0–1)	0 (0–2)	0.53	1 (0–2)	0 (0–1)	0.15
Cardiovascular risk factors *	1 (1–2)	1 (0–2)	1 (1–2)	0.57	2 (1–2.75)	1 (1–2)	0.23
Not current smoker at 3 months	91 (50.6)	14 (33.3)	77 (55.8)	0.011	11 (55)	80 (50)	0.67
Anxiety (HADS)	5.5 (4.4)	12.2 (2.1)	3.5 (2.6)	<0.001	8.8 (4)	5.1 (4.3)	<0.001
Depression (HADS)	4.1 (3.7)	6.5 (3.9)	3.4 (3.4)	<0.001	11.9 (1.7)	3.2 (2.6)	<0.001

NIHSS: National Institutes of Health Stroke Scale. * Median (P25–P75). Mann–Whitney U test.

**Table 2 brainsci-14-00521-t002:** Characteristics of the sample 24 months after stroke by diagnosis of anxiety and depression.

	Anxiety (34)	No Anxiety (130)	*p*	Depression (26)	No Depression (138)	*p*
Age	52.7 (9.4)	60.9 (10.9)	<0.001	58.2 (11.0)	59.4 (11.2)	0.61
Sex						
*Male*	20 (58.8)	107 (82.3)	0.004	21 (80.8)	106 (76.8)	0.65
*Female*	14 (41.2)	23 (17.7)		5 (19.2)	32 (23.2)	
Other addictions	11 (32.4)	33 (25.4)	0.41	8 (30.8)	36 (26.1)	0.62
Primary level education	24 (70.6)	83 (63.8)	0.46	17 (65.4)	90 (65.2)	0.98
Living with a partner	8 (23.5)	55 (42.3)	0.045	10 (38.5)	53 (38.4)	0.99
Ischaemic stroke	26 (76.5)	115 (88.5)	0.07	21 (80.8)	120 (87)	0.40
Side						
*Left*	14 (41.2)	69 (53.5)	0.24	10 (40)	73 (52.9)	0.20
*Right*	18 (52.9)	48 (37.2)		14 (56)	52 (37.7)	
*Bilateral*	2 (5.9)	12 (9.3)		1 (4)	13 (9.4)	
Insula lesion	2 (5.9)	17 (13.2)	0.23	2 (8)	17 (12.3)	0.40
Rankin at 24 months *	1 (0–2)	1 (0–2)	0.69	2 (1–2)	1 (0–1)	<0.001
NIHSS at 24 months *	0 (0–1)	0 (0–1)	0.89	1 (0–3)	0 (0–1)	0.001
Cardiovascular risk factors *	1 (1–3)	1 (1–2)	0.42	1 (1–3)	1 (1–2)	0.48
Not current smoker at 3 months	10 (29.4)	68 (52.3)	0.017	9 (34.6)	69 (50)	0.15
Anxiety (HADS) 24 months	12.7 (2.0)	3.5 (2.3)	<0.001	9.3 (4.7)	4.6 (3.9)	<0.001
Depression (HADS) 24 months	9.3 (4.3)	3.5 (3.2)	<0.001	12.3 (2.1)	3.3 (2.7)	<0.001

NIHSS: National Institutes of Health Stroke Scale. * Median (P25–P75). Mann–Whitney U test.

**Table 3 brainsci-14-00521-t003:** Independent variables associated with perceived health, evaluated at 3, 12 and 24 months in men *.

Independent Variable	B	SE	Beta	*p*	Adjusted R^2^
**At 3 months**					0.329
Depression	−1.168	0.408	−0.246	0.005	
Anxiety	−1.035	0.352	−0.239	0.004	
Rankin Scale	−3.890	1.337	−0.233	0.004	
Not current smoker	5.562	2.597	0.152	0.034	
**At 12 months**					0.482
Depression	−1.614	0.341	−0.367	<0.001	
Pain/discomfort	−6.379	1.677	−0.277	<0.001	
Rankin Scale	−4.474	1.410	−0.222	0.002	
Not current smoker	5.553	2.584	0.140	0.034	
**At 24 months**					0.371
Depression	−2.320	0.387	−0.455	<0.001	
Rankin Scale	−6.653	1.772	−0.285	<0.001	

* Multiple linear regression.

**Table 4 brainsci-14-00521-t004:** Independent variables associated with perceived health, evaluated at 3, 12 and 24 months in women *.

Independent Variable	B	SE	Beta	*p*	Adjusted R^2^
**At 3 months**					0.517
Depression	−3.627	0.577	−0.728	<0.001	
**At 12 months**					0.445
Depression	−2.298	0.592	−0.494	<0.001	
Not current smoker	17.037	5.670	0.383	0.005	
**At 24 months**					0.451
Depression	−3.970	0.718	−0.683	<0.001	

* Multiple linear regression.

**Table 5 brainsci-14-00521-t005:** Predictor variables at the evaluation 3 and 12 months after stroke of the state of perceived health at 24 months in men and women *.

Independent Variable	B	SE	Beta	*p*	Adjusted R^2^
**MEN**					
**At 3 months**					0.127
Anxiety	−1.401	0.434	−0.276	0.002	
Not current smoker	8.895	3.720	0.205	0.018	
**At 12 months**					0.357
Pain/discomfort	−9.350	1.895	−0.370	<0.001	
Rankin Scale	−7.161	1.735	−0.312	<0.001	
Not living with a partner	−7.930	3.259	−0.178	0.016	
Not current smoker	7.495	3.179	0.173	0.020	
**WOMEN**					
**At 3 months**					0.193
NIH	−8.074	2.607	−0.464	0.004	
**At 12 months**					0.229
Depression	−1.992	0.767	−0.382	0.014	
NIH	−6.896	3.154	−0.321	0.036	

* Multiple linear regression.

## Data Availability

The data underlying this article will be shared on reasonable request to the corresponding author due to their containing information that could compromise the privacy of the study participants.

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
