# Peer review of "Sex-Related Differences in Post-Stroke Anxiety, Depression and Quality of Life in a Cohort of Smokers"

_brainsci, 2024, doi:10.3390/brainsci14060521_

Round 1
Reviewer 1 Report
Comments and Suggestions for Authors
In this study, Suñer-Soler et al. investigated post-stroke mental health in smokers, revealing sex-related differences in anxiety, depression, and quality of life. The results highlighted the impact of continued smoking and stroke severity on perceived health over a 24-month period. The topic of this study is meaningful, but manuscript writing could be improved. Here are some specific comments on this study:
1. According to the instructions of Brainsci, the abstract should be a single paragraph and should follow the style of structured abstracts, but without headings (https://www.mdpi.com/journal/brainsci/instructions#preparation).
2. Line 29 “HAD”, please define the abbreviation, and the other abbreviations should be defined when they appear the first time.
3. In the introduction the authors should provide more background on relevant research on stroke anxiety. Lines 63-66, the authors should provide more detailed information.
4. Line 80 and line 135, “to the XXX 80 University Hospital in XXX with an acute stroke diagnosis (” specific information should be provided.
5. lines 101-109, different font sizes.
6. Line 284 “patients.4” different citation formats.
7. It is recommended that the authors mention and emphasize the significance of this study in the discussion part.
8. I consider the sex (male 143 and female 37) limitation in this study and that the authors should discuss it in the limitation part.
Author Response
Reviewer 1
Comments and Suggestions for Authors
In this study, Suñer-Soler et al. investigated post-stroke mental health in smokers, revealing sex-related differences in anxiety, depression, and quality of life. The results highlighted the impact of continued smoking and stroke severity on perceived health over a 24-month period. The topic of this study is meaningful, but manuscript writing could be improved.
The authors appreciate your comments and review of our manuscript.
Here are some specific comments on this study:
- According to the instructions of Brainsci, the abstract should be a single paragraph and should follow the style of structured abstracts, but without headings (https://www.mdpi.com/journal/brainsci/instructions#preparation).
Answer: The abstract has been reduced to 200 words in the format indicated.
- Line 29 “HAD”, please define the abbreviation, and the other abbreviations should be defined when they appear the first time.
Answer: HAD has been removed from the abstract and replaced with the full name and the abbreviation is now introduced in the methodology section. The rest of the abbreviations have also been revised.
- In the introduction the authors should provide more background on relevant research on stroke anxiety. Lines 63-66, the authors should provide more detailed information.
Answer: We have added more information to the introduction about anxiety in people with stroke.
- Line 80 and line 135, “to the XXX 80 University Hospital in XXX with an acute stroke diagnosis (” specific information should be provided.
Answer: The name of the centre where the research was carried out has been given.
- Lines 101-109, different font sizes.
Answer: This has been corrected and revised in the rest of the manuscript.
- Line 284 “patients.4” different citation formats.
Answer: This has been corrected
- It is recommended that the authors mention and emphasize the significance of this study in the discussion part.
Answer: We have now added more information to the discussion part about the significance and importance of the study.
- I consider the sex (male 143 and female 37) limitation in this study and that the authors should discuss it in the limitation part.
Answer: We agree with this observation and have now added this difference as a limitation and commented on it at the end of the discussion.

Reviewer 2 Report
Comments and Suggestions for Authors
A well-orchestrated document based on the search for sex/gender-based differences in aging, depression and post-stroke quality of life in smoking subjects. The data is interesting, it would probably be better if the groups were larger, and would give us more information on the different effects of stroke between men and women, and how to predict them and use different strategies. Evaluating the effect of dimorphism on these post-stroke manifestations is justified by recent evidence demonstrating that structures in the human brain exhibit sexual dimorphism. Precisely, it has been recently reported that the amygdalae are larger in males than in females and, with respect to memory formation, dimorphism occurs in terms that emotional memories in the female amygdala involve the left side (visually predominant, positive and negative). , while in males it involves the right side (negative emotional responses). Furthermore, prefrontal cortical regions exhibit increased expression of estrogen receptors, which could explain differences in decision making between the sexes. Structural neuroimaging explorations have also found that human males have reduced levels of overall cortical thickness and an increased rate of decline in cortical thickness, while females have a greater volume of white matter. Sexual variations in humans have also been found in neurotransmitter systems, including adrenergic, serotonergic, cholinergic, corticosterone, benzodiazepine, and cholecystokinin, factors largely associated with episodic memory. Notably, males produce more serotonin than females, which could influence disease states associated with serotonin dysfunction. This may influence differences in the learning process, as observed during stress, and the subsequent impact on conditioning (increased in males but inhibited in females). This association between dimorphic responses to stress is largely related to sex hormones. Studies in human and animal models of aging, cognitive decline, neurodegenerative and psychiatric diseases have shown that the latter (sex hormones) influence the permeability of the blood-brain barrier (BBB). As increased BBB permeability is hypothesized to be a major pathophysiological hallmark of neurodegenerative diseases, these findings may have broader implications for sexual dimorphism in these disease processes, although further research is essential to test and validate this association.
That said, hormone investigations should probably also be performed; in the study, women showed more significant neurological disorders in the short and long term than men
I believe to make a difference that non-smoking post-stroke subjects should also be considered, and then these more evident manifestations in women should be compared with the general clinical conditions of women and men
The work requires greater revision, trying to satisfy the points raised by implementing all the sections and final considerations, thus improving the understanding of the authors' final message
Comments on the Quality of English LanguageModerate editing of English language required
Author Response
Reviewer 2
Comments and Suggestions for Authors
A well-orchestrated document based on the search for sex/gender-based differences in aging, depression and post-stroke quality of life in smoking subjects. The data is interesting, it would probably be better if the groups were larger, and would give us more information on the different effects of stroke between men and women, and how to predict them and use different strategies. Evaluating the effect of dimorphism on these post-stroke manifestations is justified by recent evidence demonstrating that structures in the human brain exhibit sexual dimorphism. Precisely, it has been recently reported that the amygdalae are larger in males than in females and, with respect to memory formation, dimorphism occurs in terms that emotional memories in the female amygdala involve the left side (visually predominant, positive and negative), while in males it involves the right side (negative emotional responses). Furthermore, prefrontal cortical regions exhibit increased expression of estrogen receptors, which could explain differences in decision making between the sexes. Structural neuroimaging explorations have also found that human males have reduced levels of overall cortical thickness and an increased rate of decline in cortical thickness, while females have a greater volume of white matter. Sexual variations in humans have also been found in neurotransmitter systems, including adrenergic, serotonergic, cholinergic, corticosterone, benzodiazepine, and cholecystokinin, factors largely associated with episodic memory. Notably, males produce more serotonin than females, which could influence disease states associated with serotonin dysfunction. This may influence differences in the learning process, as observed during stress, and the subsequent impact on conditioning (increased in males but inhibited in females). This association between dimorphic responses to stress is largely related to sex hormones. Studies in human and animal models of aging, cognitive decline, neurodegenerative and psychiatric diseases have shown that the latter (sex hormones) influence the permeability of the blood-brain barrier (BBB). As increased BBB permeability is hypothesized to be a major pathophysiological hallmark of neurodegenerative diseases, these findings may have broader implications for sexual dimorphism in these disease processes, although further research is essential to test and validate this association.
That said, hormone investigations should probably also be performed; in the study, women showed more significant neurological disorders in the short and long term than men
I believe to make a difference that non-smoking post-stroke subjects should also be considered, and then these more evident manifestations in women should be compared with the general clinical conditions of women and men
The work requires greater revision, trying to satisfy the points raised by implementing all the sections and final considerations, thus improving the understanding of the authors' final message
Comments on the Quality of English Language: Moderate editing of English language required
Answer:
We appreciate your comments, which are extremely interesting. We would like to explain that it was not the aim of our study to include in the sample people who did not smoke before the stroke, since we are a research group with a line of research on post-stroke smoking cessation that has been going for more than a decade.
We would also like to point out that our lines of research, until now, have been to find results that are applicable in healthcare practice, especially based on the early detection of post-stroke complications, such as anxiety or depression, and that attempt to encourage affected people to modify their unhealthy behaviours in order to prevent new stroke recurrences, which is why the first year of follow-up is essential.
In the future, it would be of great interest to include in research designs variables related to hormonal levels, neurotransmitter studies, and more complex neuroimaging studies, which were not in the design of the present research. We thank you again for all your explanations, in this respect we have carried out new bibliographic searches to expand and justify our results regarding the worse mental health of women, along the lines that you have told us. As a result, we have added new references in the discussion.

Round 2
Reviewer 2 Report
Comments and Suggestions for Authors
All the suggestions have been addressed